# Mixed Eccrine Cutaneous Tumor with Folliculo–Sebaceous Differentiation: Case Report and Literature Review

**DOI:** 10.3390/medicina59081465

**Published:** 2023-08-16

**Authors:** Dimitrinka Kisova, Tihomir Dikov, Vesela Ivanova, Hristo Stoyanov, Greta Yordanova

**Affiliations:** 1Department of General and Clinical Pathology, Faculty of Medicine, Medical University Sofia, 1431 Sofia, Bulgaria; tdikov@medfac.mu-sofia.bg (T.D.); vivanova@medfac.mu-sofia.bg (V.I.); 2Department of Maxillofacial Surgery, Alexandrovska University Hospital, 1431 Sofia, Bulgaria; drhristostoyanov@gmail.com; 3Department of Orthodontics, Faculty of Dental Medicine, Medical University Sofia, 1431 Sofia, Bulgaria; g.yordanova@fdm.mu-sofia.bg

**Keywords:** skin, adnexal, tumors, eccrine mixed tumor, histopathology

## Abstract

*Background/Introduction:* Cutaneous mixed tumor is a rare benign neoplasm that exhibits a wide range of metaplastic changes and differentiation in the epithelial, myoepithelial, and stromal components, which is often confused with various other skin lesions. *Case report:* We present an unusual case of a 58-year-old woman with a mixed tumor of the upper lip, previously misdiagnosed as adnexal carcinoma on a preoperative biopsy. The excision biopsy shows a well-circumscribed lesion composed of various cells and structures featuring folliculo–sebaceous differentiation embedded in a prominent chondromyxoid stroma. The immunohistochemical study proves the various lineages of differentiation and classifies the neoplasm as the less common eccrine subtype of cutaneous mixed tumor. *Discussion:* The common embryologic origin of the folliculo–sebaceous apocrine complex leads to a great histological variety of cellular components of mixed tumors and the formation of structures that resemble established types of adnexal neoplasms, which could be a diagnostic pitfall, especially on a small incision biopsy.

## 1. Background

Cutaneous mixed tumor, previously known as chondroid syringoma, is a rare benign adnexial/appendageal tumor considered to be the cutaneous analogue of pleomorphic adenoma. The most widely reported frequency is around 0.01% of all cutaneous neoplasms showing a preference for the head and neck region. Mixed tumors usually present as solitary well circumscribed lesions in middle-aged people with slight male predominance [1].

These tumors usually exhibit a wide range of metaplastic changes and differentiation in the epithelial, the myoepithelial, and the stromal components. Sometimes these changes are so prominent as to cause misdiagnosis with other adnexal or mesenchymal neoplasms [2].

Cutaneous mixed tumors are usually further subclassified into the more common apocrine type and the rare eccrine type. The apocrine cases are generally associated with folliculo–sebaceous differentiation, where the epithelial component is composed of branching tubular structures and cystic spaces embedded in a myxoid or chondromyxoid stroma. The tubules are composed of an outer myoepithelial cell layer and an inner epithelial cell layer, sometimes showing secretion by decapitation. Solid aggregates of epithelial cells, as well as isolated polygonal and plasmacytoid (hyaline) epithelial cells, may also be seen. The eccrine variety of mixed tumors by contrast shows an epithelial component represented by small tubules lined by a single layer of uniform cuboidal, oval, or round epithelial cells, having eosinophilic or hyalinized glassy cytoplasm and vesicular nuclei, without decapitation secretion. [3] These differences in the subtypes of mixed tumors are thought to be representative of the common embryologic origin of the folliculo–sebaceous apocrine complex, sometimes also referred as complex neoplasms of the primary epithelial germ, and the eccrine counterpart in contrast should be considered as a putative eccrine lesion [3,4,5].

## 2. Case Report

A 58-year-old woman presented with a several year history of a firm mass on the upper lip measuring 10/8/7 mm that had gradually become larger over time (Figure 1). Following a dermatologist consultation, a biopsy was performed with the diagnosis of a malignant trichoepithelioma.

Later, the histologic result was revised with the conclusive diagnosis of microcystic adnexal carcinoma, and the patient was referred for surgical excision. Based on the availability of the location and the well circumscribed properties of the neoplasm it was decided on a more cosmetic approach with a consecutive plastic reconstruction of the area (Figure 2). By the second postoperative week, a clean, neat, and thin scar was formed (Figure 3).

## 3. Histopathologic Assessment

The histopathologic examination of the hematoxylin-and-eosin–stained slides revealed a well circumscribed unencapsulated lesion in the subcutis without clear connection to the epidermis or adnexal structures and scant inflammatory infiltrate. With respect to the epithelial component, the following patterns were observed: tubular structures lined by one or two layers of epithelial cells, strands/cords, single cell distribution, cribriform structures, syringomatoid tadpole-like structures, and small solid nests (Figure 4A,D). The epithelial component also showed variations in the cell appearance with the focal presence of plasmacytoid (hyaline) cells, shadow cells, physaliphorous-like cells, and intracellular vacuolization reminiscent of abortive luminal differentiation [1].

The follicular differentiation was determined by the presence of infundibular cysts (cystlike structures lined by a stratified squamous epithelium that contains a granular layer and corneocytes in lamellar pattern) (Figure 4B), basaloid metrical cells, trichohyalin granules, follicular germlike structures, and trichilemmal differentiation (palisaded pale or clear columnar cells often associated with a basal membrane, often with nuclei at the pole opposite to the basement membrane, indicating follicular differentiation toward the outer sheath at the bulb) (Figure 4C,E).

No other metaplastic or other cellular alterations involving the epithelial components were observed. The myoepithelial component consisted of strands/cords and small solid nests.

Sebaceous differentiation was determined by the presence of mature adipocytes with vacuolated cytoplasm and scalloped nuclei, some of which connected to trichilemmal cyst-like structures (Figure 5A).

No apocrine differentiation in the form of decapitation secretion was observed.

Apart from the typical myxohyaline and cartilaginous appearances of the stromal component, other variations were observed like lipomatous metaplasia consisting of small clusters of mature lipocytes, some of which contained intranuclear inclusions in the form of so-called Lochkern [1] (Figure 5).

Careful examination of the slides from the previous biopsy showed some of the same patterns: infundibular cysts, double-layered tubular structures, and trichilemmal and sebaceous elements, as well as some strands/cords of cells and clusters of mature adipocytes (Figure 6). The stromal component was mostly of the spindled cell variety, lacking the myxochondroid appearance of the excisional biopsy. These findings led us to deduct that this lesion was misdiagnosed as microcystic adnexal carcinoma for it in fact contains a lot of the hallmarks of cutaneous mixed tumor without the characteristic stomal appearance.

## 4. Immunohistochemical Study

Immunohistochemical stains were performed on tissue sections from both the incisional and excisional biopsy. CEA and EMA were performed on the first specimen showing positive staining of the luminal cells of the neoplastic tubules, with occasional staining of the secretory material (Figure 7A,B). The epithelial cells were uniformly positive for CK14/5 confirming the eccrine nature of the mixed tumor, whereas GCDF15 was negative except for a single neoplastic tubular structure (Figure 7C,D). BerEP4 showed positive areas trichoblastic differentiation, and p63 was positive in the cells showing a myoepithelial morphologic phenotype (Figure 7E,F) [6].

## 5. Discussion

Mixed tumor of the skin, arguably referred to as chondroid syringoma, is a rare and mostly benign skin lesion, comprising epithelial, myoepithelial, and mesenchymal stromal-derived elements [7]. The most commonly reported location is in the head and neck region with a frequency of less than 0.1% [7,8]. Due to the wide variety of differentiation and metaplastic changes in these lesions, it may present a diagnostic pitfall as it could mimic various skin and adnexal neoplasms. Cutaneous mixed tumor may be a diagnosis more frequent than expected, as case studies show a high risk of misdiagnosis, especially on a preoperative biopsy material, most commonly confused with other benign lesions such as dermoid or sebaceous cysts, neurofibromas, dermatofibromas, histiocytomas, pilomatricomas, and seborrheic keratosis [8,9]. The epithelial component is comprised of neoplastic structures that resemble established types of adnexal neoplasms, some of which are recognized in syringoma and apocrine hidradenoma, as well as some malignant differential diagnoses: carcinomas with a mucinous stroma, carcinosarcoma, mucoepidermoid carcinoma, and not rarely basal cell carcinoma [10,11,12]. We as well are presenting a clinical case with the misdiagnosis of microcystic adenocarcinoma on the preoperative biopsy. In a limited biopsy, lesions of this nature could represent a diagnostic pitfall by potentially mimicking a metastatic carcinoma [3]. Another potential diagnostic pitfall is the presence of physaliphorous-like cells occasioning a resemblance to the cells of a chordoma [3]. Some authors have discussed the differential diagnosis of such tumors with some mesenchymal neoplasms, including osteoma cutis and cutaneous chondroma [11,13,14]. Mixed tumors may show a prominent myoepithelial component with changes such as collagenous spherulosis, a feature originally reported in the breast; this occurrence in the salivary glands has been reported in myoepitheliomas and pleomorphic adenomas [1].

Overall, most of the available cases report a definite association between the apocrine subtype of cutaneous mixed tumor and folliculo–sebaceous differentiation [15]. Our case represents an undeniable folliculo–sebaceous differentiation with immunohistochemically verified eccrine origin of the neoplastic tubular structures (CK14/5+), which broadcasts common histogenesis of the folliculo–sebaceous–apocrine complex and the eccrine glands. This proves further investigation is needed to determine the definite progenitor cell type of mixed tumor and the usefulness of ICH for determining the tumor subtype as either eccrine or apocrine.

The most effective method for definitive diagnosis of mixed tumor is total excision and histopathologic examination. Fine-needle aspiration cytology has also been described for the diagnosis of mixed tumor, but for such a small subcutaneous lesion, this remained ineffective [8].

Two distinct cytologic components are reported to be diagnostic by FNA: a mesenchymal element with a chondroid appearance and an epithelial component. Although FNA may be useful if it is necessary to determine the pathology prior to definitive excision, histologic examination of an excised specimen remains the definitive method of diagnosis [16,17].

Although this tumor may be treated with various methods, including electrodesiccation, dermabrasion, and vaporization with argon or CO_2_ lasers, the treatment of choice is complete excision with a cuff of normal tissue in order to examine the histopathologic features [8,10]. As incidence of non-melanoma skin cancer is increasing, it is proving to be a public health problem, taking also into account the higher treatment costs, especially at the level of the face, as a more cosmetic approach is desired. Concluding the correct diagnosis is of utmost importance not only for patient prognosis, but it has also been observed that treatment costs differ according to the type of the neoplasm. [18]

The literature reviews suggest that recurrences were seen only when treatment consisted of electrodesiccation or incomplete excision. Lesions did not recur once they had been removed in toto [7]. Malignant transformation is exceptionally rare, with only a few cases reported where additionally a long term followup is recommended, as some authors even suggest adjuvant radiotherapy [19,20,21].

The heterogenicity of cutaneous mixed tumors highlights the importance of obtaining adequate tissue for histologic evaluation, as they can be confused with other skin neoplasms because of their clinically ambiguous presentations.

## Figures and Tables

**Figure 1 medicina-59-01465-f001:**
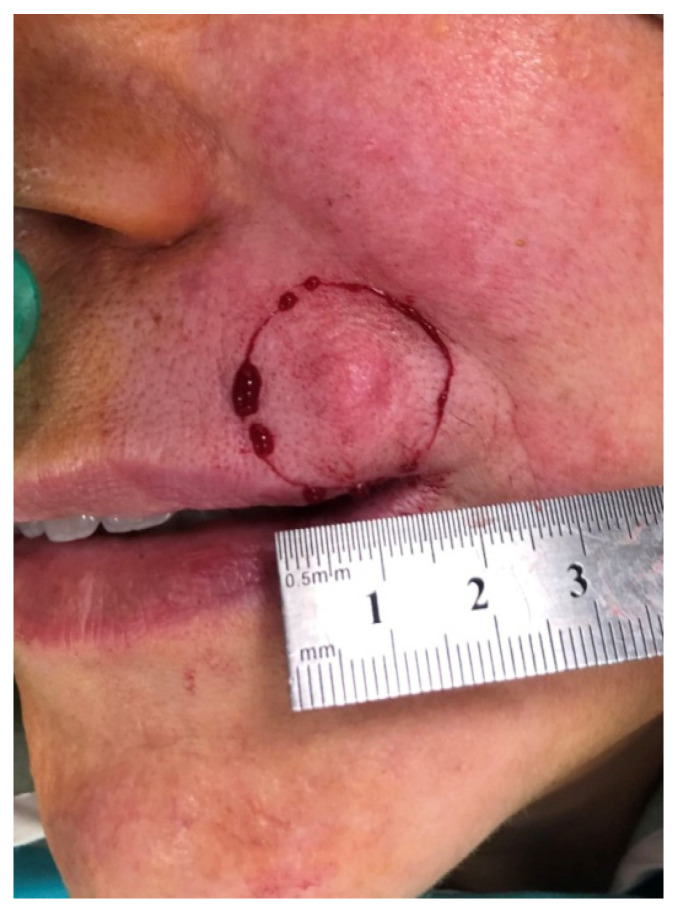
A firm nodule on the upper lip area.

**Figure 2 medicina-59-01465-f002:**
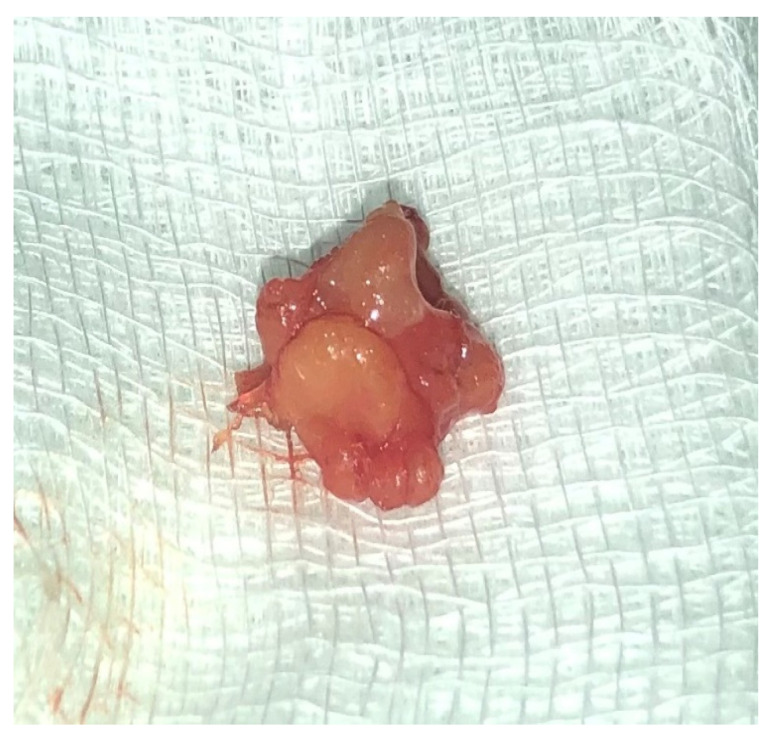
Well circumscribed subcutaneous tan lesion.

**Figure 3 medicina-59-01465-f003:**
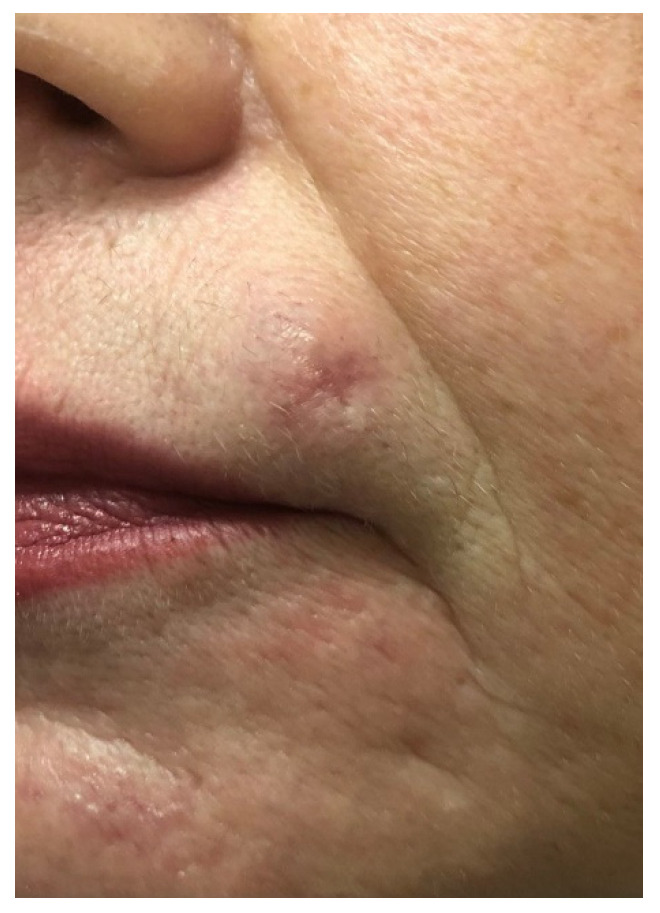
Two weeks after surgery.

**Figure 4 medicina-59-01465-f004:**
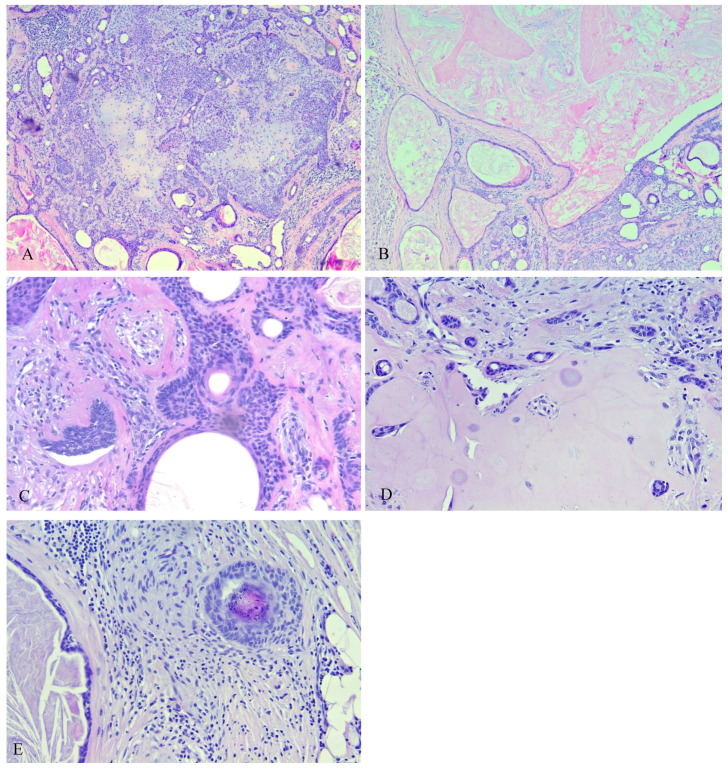
(**A**) Tubular structures and solid areas in a mixochondroid background. (**B**) Infundibular cysts in various sizes with corneocytes in lamellar pattern. (**C**) Follicular germlike structures, (small aggregations of germinative cells with peripheral palisading, with or without associated whorls of delicate collagen bundles and thin fibroblasts). (**D**) Comma/tadpole shaped ducts; syringoma like eccrine structures. (**E**) Cells resembling those of the internal epithelial sheath (trichohyalin granules).

**Figure 5 medicina-59-01465-f005:**
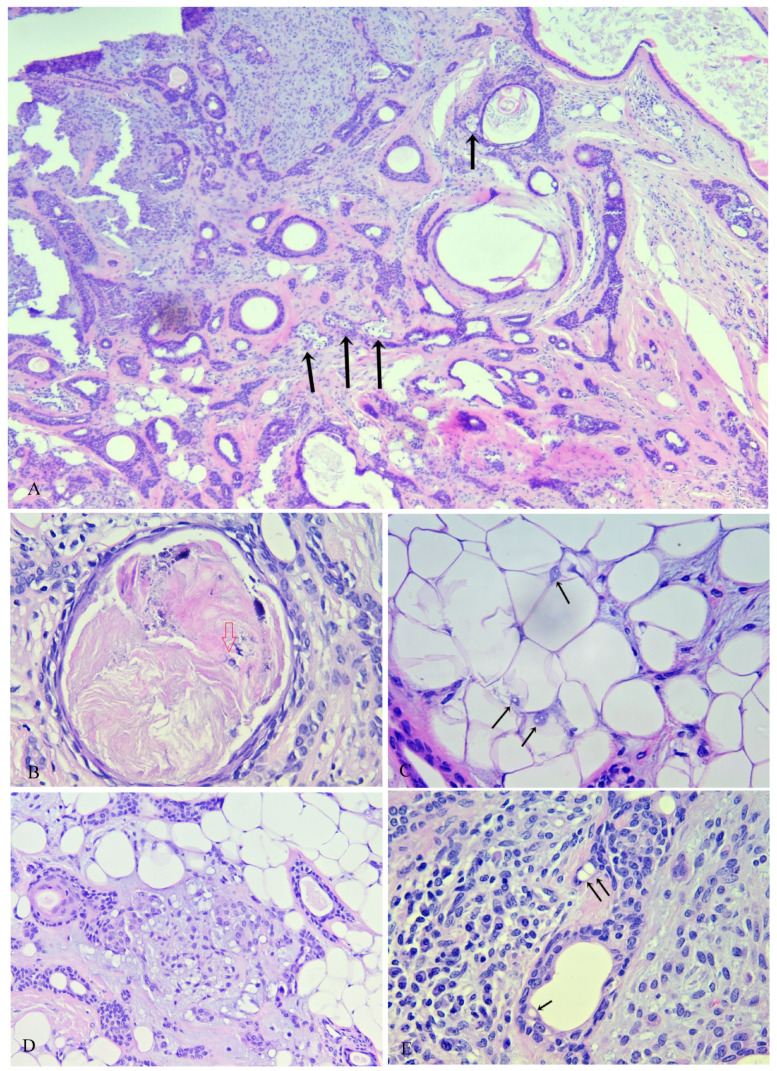
(**A**) Sebaceous gland elements (black arrows). (**B**) Shadow cells (red arrow). (**C**) Lipomatous metaplasia—mature adipocytes with intranuclear inclusions—Lochkern. (**D**) Physaliphorous-like cells. (**E**) Intracytoplasmic vacuoles.

**Figure 6 medicina-59-01465-f006:**
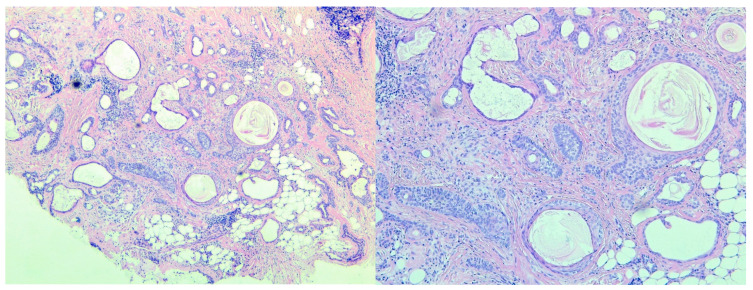
H&E stained slides from the first incisional biopsy of the patient.

**Figure 7 medicina-59-01465-f007:**
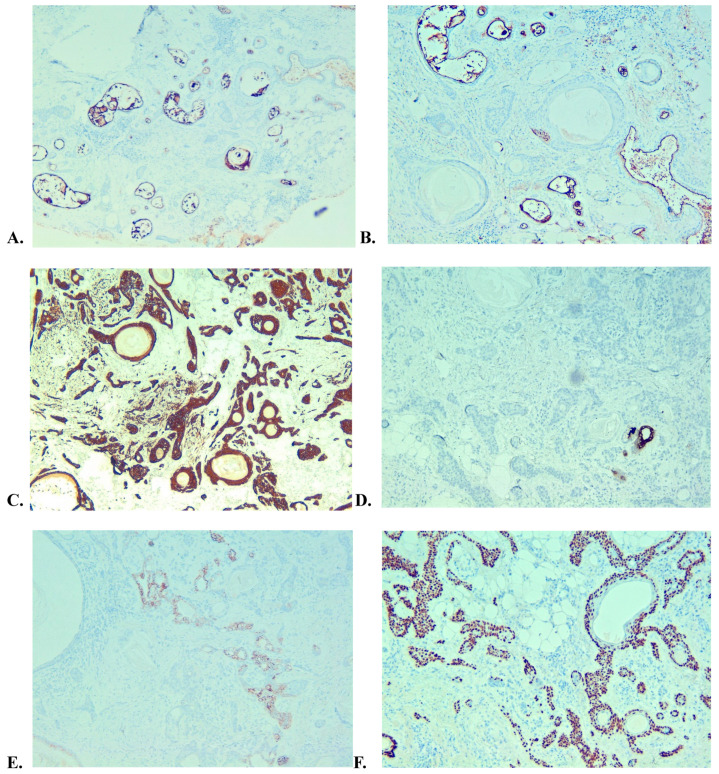
(**A**,**B**) CEA and EMA, respectively, showing positive staining in the luminal epithelial cells as well as in the secretory material but overall negativity in the neoplastic cells. (**C**) Positive expression of CK14/5 in the epithelial cells. (**D**) GCDFP15 positivity is a single neoplastic tubule. (**E**) BerEP4 positivity in a portion of the neoplastic tubules. (**F**) p63 positive staining is shown in the outer layer of the neoplastic tubules as part of the myoepithelial component.

## Data Availability

The authors confirm that the data supporting the findings of this study are available within the article.

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
