# Peer review of "Mixed Eccrine Cutaneous Tumor with Folliculo–Sebaceous Differentiation: Case Report and Literature Review"

_medicina, 2023, doi:10.3390/medicina59081465_

Round 1

Reviewer 1 Report

Dear authors,

You present a very interesting case of eccrine tumor at the level of the face.

However, there are some aspects requiring your attention in order to improve the manuscript.

Reinsert Figure 3, because the caption letters are on one page and the images on the next page.

In the discussion section mention the differential diagnosis with other non-melanoma skin cancer at the level of the face. Reference this to newer articles from MDPI platform like Faur, C.I.; Moldovan, M.A.; Văleanu, M.; Rotar, H.; Filip, L.; Roman, R.C. The Prevalence and Treatment Costs of Non-Melanoma Skin Cancer in Cluj-Napoca Maxillofacial Center. Medicina 202359, 220. https://doi.org/10.3390/medicina59020220.

At the end of the manuscript insert the sections regarding the author contributions, and the ethical statements.

Looking forward to receiving the improved version of your manuscript.

Author Response

Dear Reviewer, 

Thank you for your consideration and valuable input regarding this manuscript. 

Indeed I have had some issues with Word formatting which is why I also included a pdf version of the manuscript that could be used for reference. Nevertheless I have tried again to correct the issue, so hopefully it will work better this time. 

The article you referenced will bring an interesting and novel point to the discussion which I think will contribute  to the final well-rounded outlook of the manuscript. I will make sure to reference it. 

Thank you as well for bringing to my attention the need to include sections regarding the author contributions, and the ethical statements.

I hope you enjoy the final version better. 

Best regards. 

Reviewer 2 Report

Dear Authors,

You present an interesting case of eccrine tumor.

There are some aspects that could improve the manuscript.

Please insert postoperative image with the patient.

Did you perform any imaging study? Like a CT scan? Because the initial suspicion was of a malignant tumor.

Pay attention at the end of the manuscript to include the sections about author contribution, funding, ethics, and others.

Wish you good luck with this manuscript.

Author Response

Dear Reviewer, 

Thank you for your consideration and expert input regarding this manuscript. 

Inserting a postoperative picture of the patient is a great idea as it gives the whole case a sense of completeness, so I look forward to incorporating the image as it is readily available to us. 

Even though the lesion was initially suspected to be malignant the easily accessible area prompted a quick excision of the tumor. Later on as we concluded the benign character of the lesion there was no need for a CT scan. 

Thank you as well for bringing to my attention the need to include sections about ethics statements, author contribution and funding. 

I hope you enjoy the revised version of the manuscript better.

Best regards.

Reviewer 3 Report

This is an interesting case report with a concomitant literature review on Mixed eccrine cutaneous tumor with follicular-sebaceous differentiation. The article correctly points out the difficulties in achieving a correct diagnosis of mixed tumors of the skin. Misdiagnosis may be a cause of bad management of this particular group of tumors, so I feel that the discussion on this subject is very important. I really liked the part regarding histology and immune-histochemistry, which might be very helpful in differential diagnosis of complex cases.

Author Response

Dear Reviewer, 

I greatly appreciate your opinion and consideration of this manuscript. 

I do hope it will contribute in a wholesome manner in the understanding, diagnosis and treatment of this neoplasm, as it seems to be more common than expected. 

I look forward for you to enjoy the revised version of the manuscript. Any input would be kindly appreciated and heeded. 

Best regards. 

Round 2

Reviewer 1 Report

Dear Authors,

I am glad you answered all the comments from the reviewers.